



# Exploring the relationship between temperature forecast errors and Earth system variables

Melissa Ruiz–Vásquez[1,2], Sungmin O[3], Alexander Brenning[2], Randal D. Koster[4], Gianpaolo Balsamo[5], Ulrich Weber[1], Gabriele Arduini[5], Ana Bastos[1], Markus Reichstein[1], and René Orth[1]

[1]Department of Biogeochemical Integration, Max Planck Institute for Biogeochemistry, Jena, Germany
[2]Friedrich Schiller University Jena, Department of Geography, Jena, Germany
[3]Department of Climate and Energy System Engineering, Ewha Womans University, Seoul, South Korea
[4]Global Modeling and Assimilation Office, NASA Goddard Space Flight Center, Greenbelt, MD, USA
[5]Research Department, European Centre for Medium Range Weather Forecasts, Reading, Great Britain

**Correspondence:** Melissa Ruiz–Vásquez (mruiz@bgc-jena.mpg.de)

**Abstract.**

Accurate subseasonal weather forecasts, from two weeks up to a season, can help reduce costs and impacts related to weather and corresponding extremes. The quality of weather forecasts has improved considerably in recent decades as models represent more details of physical processes, and they benefit from assimilating comprehensive Earth observation data as well
as increasing computing power. However, with ever–growing model complexity it becomes increasingly difficult to pinpoint weaknesses in the forecast models' process representations which is key to improve forecast accuracy. In this study, we use a comprehensive set of observation–based ecological, hydrological and meteorological variables to study their potential for explaining temperature forecast errors at the weekly time scale. For this purpose, we compute Spearman correlations between each considered variable and the forecast error obtained from the ECMWF subseasonal–to–seasonal (S2S) reforecasts at lead
times of 1–6 weeks. This is done across the globe for the time period 2001–2017. The results show that temperature forecast errors globally are most strongly related with climate–related variables such as surface solar radiation and precipitation, which highlights the difficulties of the model to accurately capture the evolution of the climate–related variables during the forecasting period. At the same time, we find particular regions in which other variables are more strongly related to forecast errors. For instance, in central Europe, eastern North America and southeastern Asia, vegetation greenness and soil moisture are relevant,
while in western South America and central North America, circulation–related variables such as surface pressure relate more strongly with forecast errors. Overall, the identified relationships between forecast errors and independent Earth observations reveal promising variables on which future forecasting system development could focus by specifically considering related process representations and data assimilation.

## 1   Introduction

Forecasts at the subseasonal–to–seasonal time scale (S2S) have received growing attention in recent years (Kirtman et al., 2014; Vitart et al., 2017; White et al., 2017). Accurate predictions at the S2S are helpful, for instance, to anticipate extreme





events like droughts and floods up to two weeks ahead (Bauer et al., 2015; Mariotti et al., 2018; Vitart and Robertson, 2018; Pegion et al., 2019; Pendergrass et al., 2020) and to optimize resource management (Robertson et al., 2015; White et al., 2017). The S2S bridges the gap between weather forecasts for the coming two weeks and seasonal climate predictions (Vitart, 2014; Robertson et al., 2015). However, it is particularly challenging to predict because the lead time is too long for the initial atmospheric conditions to provide useful information, and too short for some slowly varying Earth system components, such as the ocean, to effectively inform the forecasts (Vitart, 2014; Mariotti et al., 2018).

The chaotic nonlinear interactions between Earth system components and the uncertainties in the initial boundary conditions play a role in forecast errors (Newman et al., 2003). Forecasting systems need to capture all possible Earth system sources of subseasonal predictability to minimize these errors (Vitart and Robertson, 2018; De Andrade et al., 2019). The assimilation of climate variables, such as radiation and precipitation, and hence the representation of energy and water supply at the Earth's surface, is relevant to inform the weather forecast model to make accurate predictions of the evolution of temperature (Miller et al., 2021). This requires a global and dense observational network of respective measurements and/or regular satellite observations as a basis for an adequate data assimilation.

Related with climate variables, recent studies have identified potential circulation sources of S2S predictability such as Madden Julian Oscillation (MJO), El Niño Southern Oscillation (ENSO) and North Atlantic Oscillation (NAO). Atmospheric circulation patterns influence synoptic weather regimes and high/low pressure systems, which in turn modulate large–scale weather several weeks ahead (Büeler et al., 2021; Falkena et al., 2022).

Also, the land surface initial conditions (e.g. soil moisture, vegetation states, snow cover) contribute to subseasonal predictability through the memory, *i.e.* persistence in time, of related quantities, particularly in the case of soil moisture (Koster et al., 2010b, a; Saha et al., 2014; Mariotti et al., 2020; Kim et al., 2021; Meehl et al., 2021). Soil moisture can affect the partitioning of incoming radiation energy to sensible and latent heat fluxes, and therefore the near–surface temperature and humidity (Seneviratne et al., 2010). Soil moisture anomalies that are present and known at forecast initialization can persist for several weeks (Orth and Seneviratne, 2012; Shin et al., 2020), therefore an accurate representation of the soil moisture effect on surface heat fluxes supports weather forecast skill, particularly in water–limited regions (Miralles et al., 2012; Dirmeyer and Halder, 2017).

In contrast to soil moisture, the potential of vegetation phenology in improving weather forecast skills is not yet fully exploited, and in most forecasting systems, only the seasonal cycle is prescribed (Boussetta et al., 2013; Balsamo et al., 2018). Vegetation phenology (in terms of e.g. greenness and photosynthesis) affects surface heat fluxes through evapotranspiration, and exhibits memory. Some studies have outlined the potential of taking into account initial vegetation anomalies in the forecasting systems (Boussetta et al., 2015; Koster and Walker, 2015; Nogueira et al., 2020). They reveal the potential of the vegetation state in predicting hydrometeorological variables with high accuracy, especially during extreme events such as droughts and heat waves, permitting mitigation of their impacts (Meng et al., 2014; Albergel et al., 2019; Miralles et al.).

Overall, these potential sources of forecast skill at the subseasonal time scale can vary in strength across space and time, and they are utilized to different degrees in current forecasting systems. Therefore, it is not obvious which is the most promising



source of forecast skill to further exploit in forecasting system development. To provide guidance in this context, we study the temporal co–variability of errors of near–surface temperature forecasts with a comprehensive dataset of observation–based Earth system variables representing potentially under–exploited sources of forecast skill at the subseasonal time scale. With this setup, our goal is to reveal *where* (across regions) and *when* (across seasons) climate, circulation and land surface variables can explain forecast error variability.

## 2 Data and methods

Our methodology is summarized in Fig. 1, and described in more detail in the following subsections.

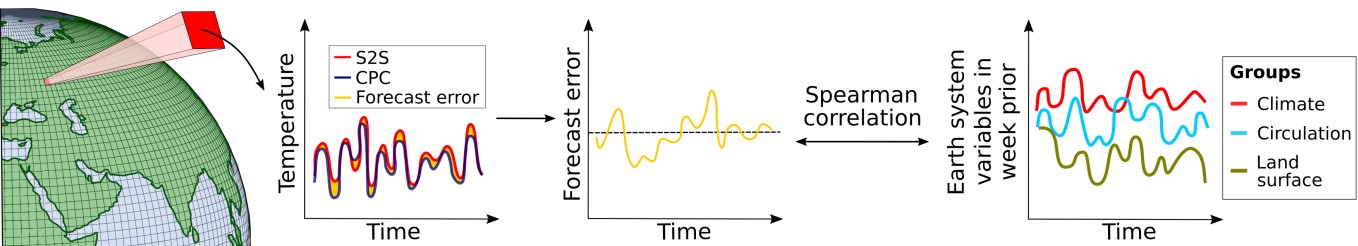

**Figure 1.** Schematic summary of the workflow. Within each grid cell, we calculate weekly averages of temperature forecast errors (lead times 1–6 weeks) and then compute their Spearman correlations with Earth system variables averaged across the week prior to the forecast start.

### 2.1 Data

Our analysis focuses on S2S reforecasts of 2m air temperatures from the European Centre for Medium–range Weather Forecasts (Vitart et al., 2017), hereafter referred to as forecast. We consider the global land area and the time period 2001–2017. Each forecast is run until a lead time of 46 days. We use forecasts generated in 2020 with two model versions: CY46R1 (all forecasts from 4 February to 29 June of each year) and CY47R1 (all forecasts from 30 June to 3 February of each year) (accessed on 3 February 2021 from https://apps.ecmwf.int/datasets/data/s2s/). The forecasts are forced with atmospheric data from the ERA5 reanalysis (Hersbach et al., 2020) and therefore provide a consistent model initialization.

All analyses in this study are performed on a weekly scale given the temporal resolution of the employed Earth system datasets and to minimize the effect of synoptic weather variability, following previous studies (Orth and Seneviratne, 2014; Koster et al., 2020). Consequently, the daily forecasts are averaged over the first, second, third, fourth, fifth and sixth weeks of the forecasts. In order to estimate errors in the temperature forecasts we compare them with respective data from the Climate Prediction Center (CPC) (accessed on 21 December 2021 from https://psl.noaa.gov/data/gridded/data.cpc.globaltemp.html). These data are model–independent and derived through interpolation of station observations. We calculate daily mean surface air temperatures as the average of each day's maximum and minimum temperatures, as recommended on the CPC website.



To assess the relationships of temperature forecast errors with multiple Earth system variables, we use the comprehensive set of data described in Table 1. A total of 21 variables are selected based on (*i*) literature about sources of predictability in seasonal forecasts and (*ii*) physical processes and quantities which can affect surface air temperature. According to the Earth

system component they are associated with, we classify these variables into three groups: climate, circulation, and land surface.

Table 1: Earth system data description

| Group | Variable | Abbrev. | Source | Reference |
|---|---|---|---|---|
| Climate | Surface solar radiation downwards | ssrd | ERA5 (reanalysis) | Hersbach et al. (2020) |
| | Surface thermal radiation downwards | strd | | |
| | Sea surface temperature at a grid cell located at the same latitude and 10° eastward of the coastline of the respective grid cell | sst1 | | |
| | Sea surface temperature at a grid cell located at the same latitude and 10° westward of the coastline of the respective grid cell | sst2 | | |
| | Specific humidity | q | | |
| | Total precipitation | tp | GPCP V3.1 (interpolation of station data) | Huffman et al. (2021) |
| Circulation | Wind speed | wind | ERA5 (reanalysis) | Hersbach et al. (2020) |
| | Surface pressure differences between 5° north and 5° south of the respective grid cell | sp diff meridional | | |
| | Surface pressure differences between 5° east and 5° west of the respective grid cell | sp diff zonal | | |
| | El Niño Southern Oscillation based on El Niño 3.4 index (anomalies only) | ENSO | World Meteorological Organization (interpolation of observations with modeling and data assimilation systems) | Trenberth (1997) |



| | | | | |
|---|---|---|---|---|
| | Madden Julian Oscillation (anomalies only) | MJO | Australian Government Bureau of Meteorology Organization (interpolation of observations with modeling and data assimilation systems) | Wheeler and Hendon (2004) |
| | North Atlantic Oscillation (anomalies only) | NAO | NCEP/NCAR (interpolation of observations with modeling and data assimilation systems) | Van den Dool et al. (2000) |
| | Pacific North American pattern (anomalies only) | PNA | | |
| | Antarctic Oscillation (anomalies only) | AAO | | Mo (2000) |
| | Arctic Oscillation (anomalies only) | AO | | Higgins et al. (2000) |
| Land surface | Snow cover fraction | snow | MODIS V6 (satellite-based data) | Hall and Riggs |
| | Enhanced Vegetation Index | EVI | | Didan |
| | Albedo | albedo | | Schaaf and Wang. |
| | Evaporative fraction | ef | FLUXCOM (up-scaled in–situ observations) | Jung et al. (2019) |
| | Surface soil moisture (0–10 cm) | sm surf | SoMo.ml (upscaled in–situ observations) | O and Orth (2021) |
| | Sub–surface soil moisture (10–50 cm) | sm deep | | |



All datasets cover the entire globe and are linearly aggregated to a spatial resolution of 0.5° x 0.5° in case this is not their native resolution, following a 2–dimensional Piecewise linear interpolation method (Barber et al., 1996). In addition to considering absolute values of all variables listed in Table 1, we also consider their anomalies in our analysis of relationships

with temperature forecast errors, except for the circulation indices ENSO, MJO, NAO, AAO, AO and PNA which already come as anomalies. To compute anomalies we (*i*) subtract the long–term trend from each variable's time series, which we infer through a Lowess smoothing filter fit, and (*ii*) we remove the mean seasonal cycle, calculated at weekly time–steps. Therefore, our final set of Earth system variables contains 36 variables (15 variables with both absolute values and anomalies, plus 6 circulation indices that are already anomalies).

We furthermore filter the vegetation–related Earth system variables in the land surface group (enhanced vegetation index (EVI), albedo, evaporative fraction) to include only sufficiently active vegetation (and hence evapotranspiration) that may physically affect temperatures(Seneviratne et al., 2010; Miralles et al., 2012). For this purpose, we do not consider grid cells nor weeks with EVI lower than 0.3 or temperatures below 10°C.

## 2.2 Forecast error

We compute the weekly temperature forecast errors using an unbiased difference metric (Yu et al., 2006; McDonnell et al., 2018) between ECMWF–S2S temperature forecasts and CPC observational reference values, as shown in Eq. (1). Thereby, at each grid cell, we (*i*) compute the average temperature of each year for both datasets, respectively; (*ii*) subtract these averages from every weekly value in the same year of the corresponding dataset; (*iii*) compute the difference between each value of forecast temperature and the corresponding reference temperature.

$$error_i = (T_{i,for} - \overline{T}_{for}) - (T_{i,ref} - \overline{T}_{ref}) \tag{1}$$

where $error_i$ is the forecast error in the *i*th week, $\overline{T}_{for}$ and $\overline{T}_{ref}$ are the annual average temperature values for the respective year from the forecast and reference datasets, respectively, $T_{i,for}$ and $T_{i,ref}$ are the forecast and reference temperature values respectively in the *i*th week.

The contributing stations of the reference temperature dataset are not uniformly distributed across the globe (Fig. A1). In

our analysis, we omit grid cells without nearby stations as they can potentially exhibit larger interpolation errors. This way, we only consider grid cells with at least one temperature station located within the grid cell or in one of its eight neighboring grid cells. We assess and analyze the temperature forecast errors separately for each season, December–January–February (DJF), March–April–May (MAM), June–July–August (JJA) and September–October–November (SON), such that for each grid cell and season we consider weekly data across 3 months per season and 17 years, resulting in approximately 220 weeks per season.



## 2.3 Relating temperature forecast errors to Earth system variable dynamics

Our hypothesis is that a high temporal correlation between temperature forecast errors and any of the Earth system variables (Table 1) would indicate that the Earth system process represented by the respective variable is not yet sufficiently exploited by the forecasting system. We use the Spearman rank correlation coefficient (Wilks, 2011) to measure the strength of the relationship between forecast errors and each considered Earth system variable. This metric is chosen to account for potentially non–linear relationships. It is calculated with Earth system variables averaged over the week prior to the forecast initialization and temperature forecast errors at lead times between 1 to 6 weeks, respectively. This way, we calculate correlation values for each of the 36 considered variables and for each weekly lead time, grid cell and season. Finally we determine the highest correlation among them (in absolute terms), which indicates the most relevant Earth system variable and consequently the most promising information source to further improve temperature forecasts. Our interpretation focuses on the relevance of variables at the level of the aforementioned groups (climate, circulation, land surface).

We only consider correlations which are significant at the 5% level. To infer significance in the light of the multiple testing at global scale (36 correlation values for the 36 Earth system variables across the globe), we perform the Benjamini–Hochberg procedure (Benjamini and Hochberg, 1995) to ensure control of the false discovery rate (Farcomeni, 2008; Cortés et al., 2020) by applying a correspondingly lower significance level. Additionally, we randomly permute the time series of each Earth system variable and compute their individual Spearman correlation values with forecast errors to compare their magnitude with our results.

Even though we analyze all weekly lead times (1 to 6), we focus on the lead time week 3 throughout most of the results section because this lead time represents well the subseasonal time scale (Brunet et al., 2010; Vitart et al., 2017; Pegion et al., 2019), which is less understood than the weather and climate scales (White et al., 2017). Also, the results after lead time week 3 do not vary substantially.

## 2.4 Case study: forecast errors in focus regions

We select six focus regions across all continents to further analyze the relationship between the temperature forecast errors and the respective most relevant Earth system variable. The regions are named according to the continent where they are located: Africa (AF), Asia (AS), Australia (AU), Europe (EU), North America (NA) and South America (SA). We choose the regions to reflect clusters of grid cells with the same determined most important variable, and to represent different continents and Earth system variable groups. Table 2 displays the specific location of the six regions and their land cover classification from the MCD12C1 MODIS/Terra+Aqua land cover type dataset V006 (Friedl and Sulla-Menashe).

Table 2: Focus regions location and description

| Region | Latitude range [°] | Longitude range [°] | Location | Sominant land cover |
|---|---|---|---|---|





| | | | | |
|---|---|---|---|---|
| AF | [-22.25, -24.75] | [28.75, 32.25] | Eastern Botswana, Southwestern Mozambique and Northeastern South Africa | Grasslands |
| AS | [26.75, 22.75] | [113.75, 119.25] | Southeastern China (Hunan, Fujian) | Mixed forest, open shrublands and woody savannas |
| AU | [-29.25, -32.75] | [147.75, 150.75] | Southeastern Australia (New South Wales) | Evergreen broadleaf forests, grasslands and woody savannas |
| EU | [49.75, 46.25] | [3.25, 7.75] | Northern France, Belgium, Southern Netherlands and Western Germany | Croplands, open shrublands, urban and built-up |
| NA | [41.25, 38.25] | [-105.25, -100.75] | Center United Stated (Wyoming, Nebraska, Kansas and Denver) | Grasslands and croplands |
| SA | [-5.75, -10.75] | [-40.75, -36.75] | Eastern Brazil (State of Ceará, State of Rio Grande do Norte, State of Sergipe, State of Bahia) | Grasslands and savannas |

## 2.5 Potential to improve temperature forecasts at the S2S time scale

Finally, we compute a measure of the potential to improve temperature forecasts in week 3 based on percentiles of (*i*) forecast errors and (*ii*) Spearman correlations of these errors with the most relevant Earth system variable identified in Sect. 3.2. Regions that exceed the 90th percentile for both quantities (forecast error and correlation values) have relatively large errors which at the same time can be attributed to an Earth system variable such that we determine this as the highest potential for improvement. For medium and low potential the grid cell values need to exceed the 70th and 50th percentiles of all global grid

cells, respectively.



## 3 Results and discussion

### 3.1 Forecast error variability

At first we analyze the temperature forecast errors as reflected by the unbiased differences between the forecasted temperature and the reference data (Fig. 2). The largest mean seasonal errors occur during DJF and JJA, with negative and positive biases

in different regions, respectively. Therefore, the smaller errors in the transition seasons MAM and SON may simply reflect a transition between these positive and negative extremes. We suggest that the larger biases in predicting temperatures during the warmest and coldest seasons (JJA and DJF) are a consequence of imperfect model physics, initial conditions and boundary conditions (Durai and Bhradwaj, 2014).

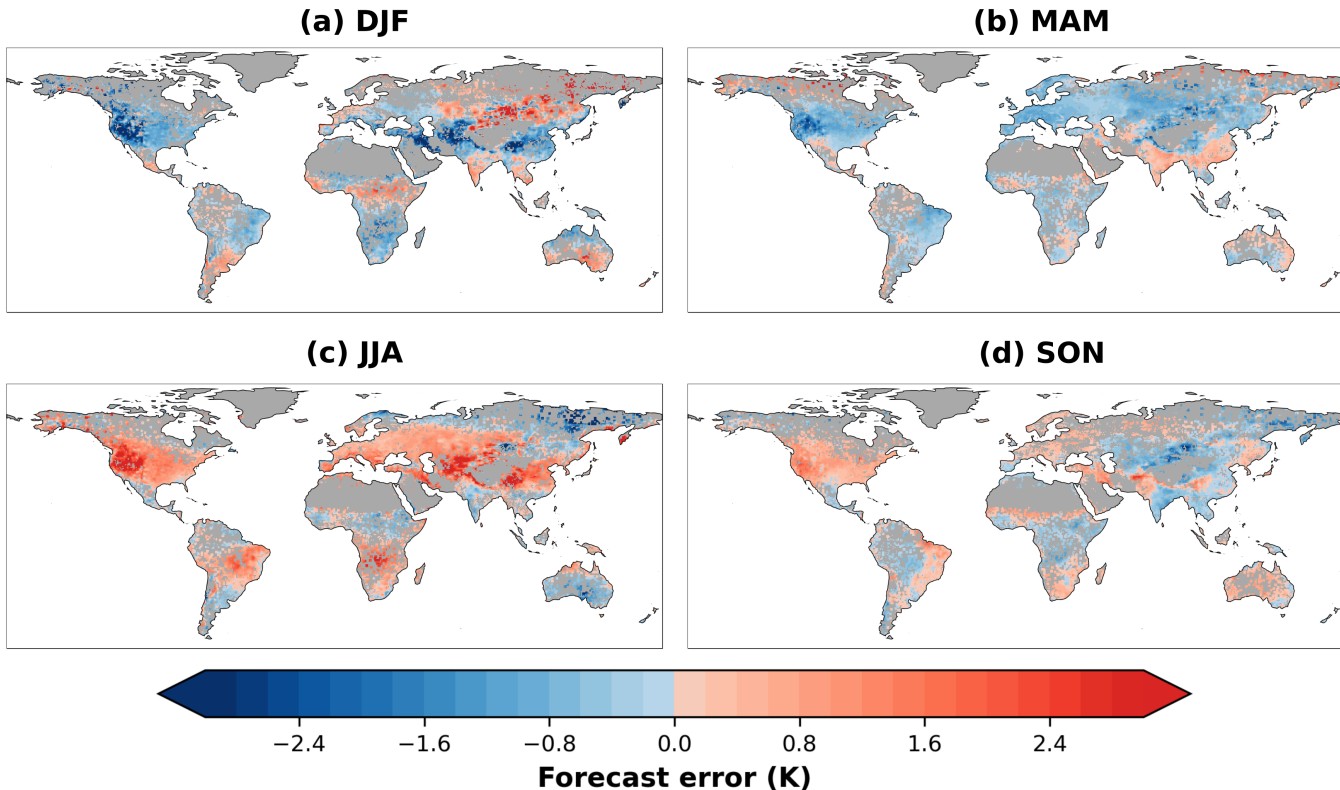

**Figure 2.** Seasonal averages (DJF, MAM, JJA, SON) of temperature forecast error. Regions with gray color are masked out of the entire analysis due to: (*i*) low observational data density (Fig. A1) or (*ii*) the Benjamini–Hochberg procedure (Sect. 2.3).

We find large errors in Asia around the Himalayas and in western North America. Dutra et al. (2021) found similar patterns

in temperature forecast error patterns for the Northern hemisphere, which they linked with the imperfect model representations of snow and soil freezing. In the tropics, the biases are smaller than in the extratropics, probably because the predictability in





tropical areas is provided mainly by slowly–varying forcings such as ENSO, while extratropic predictability is mostly derived from synoptic–scale atmospheric dynamics (Jiang et al., 2017; De Andrade et al., 2019; Judt, 2020).

The largest biases occur mainly around grid cells with low observational station density (as seen in Balsamo et al. (2018) and mountainous regions (Fig. A2). We attribute this to three possible reasons: (*i*) difficulties in representing hydrometeorological processes such as freezing and thawing in those regions (Hagedorn et al., 2008), (*ii*) the misrepresentation of mountain topography in the land surface model (Bento et al., 2017), and (*iii*) low observational station density implies greater uncertainty in the interpolation of the reference temperature dataset.

## 3.2   Relationships between temperature forecast errors and Earth system variables

Focusing on the Earth system variables with the strongest correlation with temperature forecast errors within a grid cell, we find that climate variables are overall most relevant at the global scale (Fig. 3). Variables in the climate group are also the most studied variables in terms of improving temperature forecast skill. For example, an inaccurate SST representation in climate models is associated with ENSO biases, which in turn affect temperature and precipitation on land (Kim et al., 2017; Ehsan et al., 2021; Liu et al., 2021). The most relevant individual variables for this group are precipitation and surface solar radiation

(Table 3), which control the land–atmosphere interactions and specifically surface temperature through their influence on the water and energy balances, respectively. Since the forecasts are initialized with data from the model–based ERA5 reanalysis, errors in forcing data (e.g. precipitation) can propagate to temperature forecasts with time.

[h!]

Table 3: The three most important variables for each group of variables across seasons. The first number in parenthesis is the absolute spatial average correlation of the variable with temperature forecast error. The second number in parenthesis is the variable's frequency of occurrence as the most relevant one.

|  | DJF | MAM | JJA | SON |
|---|---|---|---|---|
| Climate | anom ssrd (0.27) (4.7%) | anom ssrd (0.30) (3.5%) | anom ssrd (0.30) (9.2%) | anom ssrd (0.31) (4.2%) |
|  | abs tp (0.27) (4.1%) | anom tp (0.27) (1.5%) | abs tp (0.28) (4.2%) | abs tp (0.28) (5.0%) |
|  | anom tp (0.24) (2.0%) | abs tp (0.26) (4.2%) | abs ssrd (0.27) (13.7%) | anom tp (0.27) (2.0%) |
| Circulation | AO (0.18) (1.5%) | PNA (0.20) (0.2%) | anom sp diff meridional (0.19) (1.1%) | MJO (0.19) (0.1%) |
|  | MJO (0.16) (0.6%) | AAO (0.14) (0.1%) | (PNA) (0.11) (0.2%) | PNA (0.19) (0.9%) |
|  | abs sp diff meridional (0.09) (5.6%) | ENSO (0.13) (0.5%) | anom sp diff zonal (0.10) (1.4%) | NAO (0.17) (0.4%) |





| Land surface | anom sm1 (0.27) (3.9%) | abs sm1 (0.29) (15.0%) | anom sm1 (0.30) (4.4%) | abs sm1 (0.31) (8.7%) |
| --- | --- | --- | --- | --- |
| | abs sm1 (0.21) (4.8%) | anom sm1 (0.26) (1.5%) | abs sm1 (0.27) (8.1%) | anom sm1 (0.27) (3.1%) |
| | abs snow (0.18) (2.8%) | anom sm deep (0.25) (0.5%) | anom sm deep (0.25) (1.4%) | anom sm deep (0.25) (1.4%) |

The second most relevant group is the land surface, dominated by soil moisture as the most important variable within this group. We find that the soil moisture, especially the moisture in the uppermost layer (Table 3), is important in the Northern hemisphere during MAM and JJA (Fig. 3). Soil moisture plays a key role in the exchange of heat and water between both land and atmosphere and is one of the most studied land surface variables as a source of subseasonal predictability (Seneviratne et al., 2010; Guo et al., 2011; Orth and Seneviratne, 2014; Koster et al., 2017, 2020). Furthermore, soil moisture affects temperature forecasts particularly through its profound memory characteristics, which effectively project dry or wet anomalies 180 into the future (Koster and Suarez, 2001; Orth and Seneviratne, 2012; Galarneau and Zeng, 2020). As a result, they can affect temperature forecast quality in regions with strong land–atmosphere coupling (Koster et al., 2004; Orth, 2021). Many recent model developments are targeted at an improved representation of soil moisture dynamics and soil moisture data assimilation, and given our results, this is a promising avenue towards reducing biases in temperature estimates (Koster et al., 2011; Albergel et al., 2013; Dirmeyer and Halder, 2017; Dirmeyer et al., 2018).

Along with soil moisture, vegetation plays a primary role in land–atmosphere interactions and associated heat and water exchanges. The vegetation–related variables from the land surface group (EVI and EF) are important in some regions of South America and Africa (Fig. A4). These regions were also identified in a similar study by Boussetta et al. (2015) as regions where vegetation information, in terms of LAI and albedo anomalies, had a positive impact on the temperature forecast quality. Even though the role of vegetation information on temperature forecast skill has not been as extensively addressed as that of other 190 variables, some studies have highlighted its positive impacts in reducing forecast errors (Boussetta et al., 2013; Koster and Walker, 2015; Nogueira et al., 2020).

We find that variables from the circulation group are the least relevant for explaining temperature forecast errors globally, their importance is confined to small regions scattered across the world. The most relevant variables within this group are the large scale circulation indices NAO and MJO and the zonal and meridional surface pressure differences (Table 3), which de- 195 termine large–scale advection of air masses and hence the spatial temperature fields. A case study on this matter was presented by Grams et al. (2018). They found a connection between a strong temperature bias over Europe, predicted with the ECMWF's Integrated Forecasting System, and the misrepresentation in the onset of a blocking regime. This misrepresentation generated a chain reaction in simulated latent heat release that amplified and propagated the forecast error to a larger region. Such a



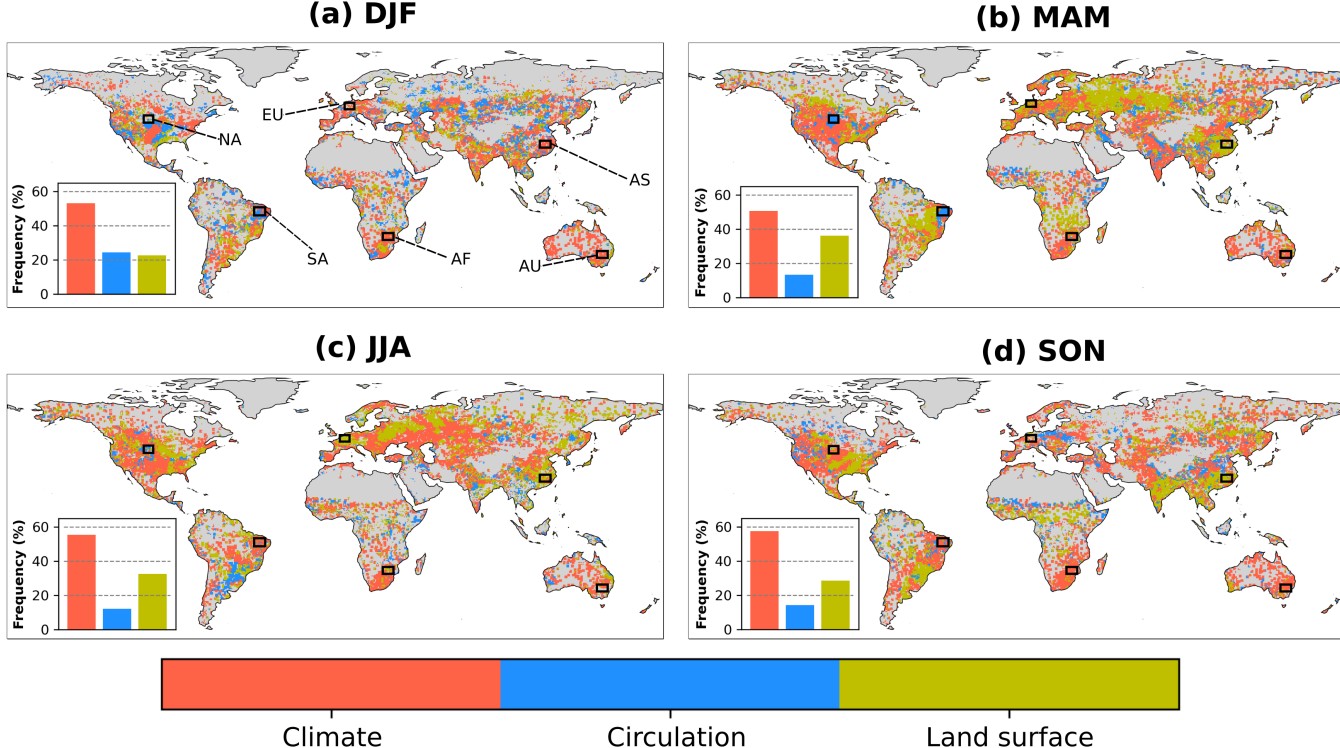

**Figure 3.** Groups of Earth system variables to which the variable with the strongest correlation with temperature forecast error belongs (based on Fig. A4). Results shown for each season (DJF, MAM, JJA, SON). The regions outlined in black are discussed in detail in Sect. 3.3: Africa (AF), Asia (AS), Australia (AU), Europe (EU), North America (NA) and South America (SA). The inset barplots represent the percentage of grid cells where variables from each group are most related to temperature forecast errors.

finding reveals the potential of the assimilation of circulation information to reduce forecast errors and uncertainties (Lei and
Anderson, 2014; Smith et al., 2020).

The overall picture of the importance of each group does not substantially change with lead time (Fig. A5). Around 23% of the grid cells show (absolute) correlations higher than 0.3 (Fig. 4). These significant correlations indicate promising areas in which temperature forecasts could be improved by incorporating information from the identified Earth system variables. The highest correlations (>0.6) and hence most promising regions for improvement are found in eastern South America, southeast
Asia and central Africa during MAM. It is noteworthy that southeast Asia also shows high temperature forecast errors (Fig. 2). It should, however, be noted that despite being significant after correction for multiple testing, and also being substantially different from correlations based on non–sense data (Fig. A3), these correlations from the present exploratory analysis do not directly indicate the extent to which the forecast error could potentially be improved based on these variables.




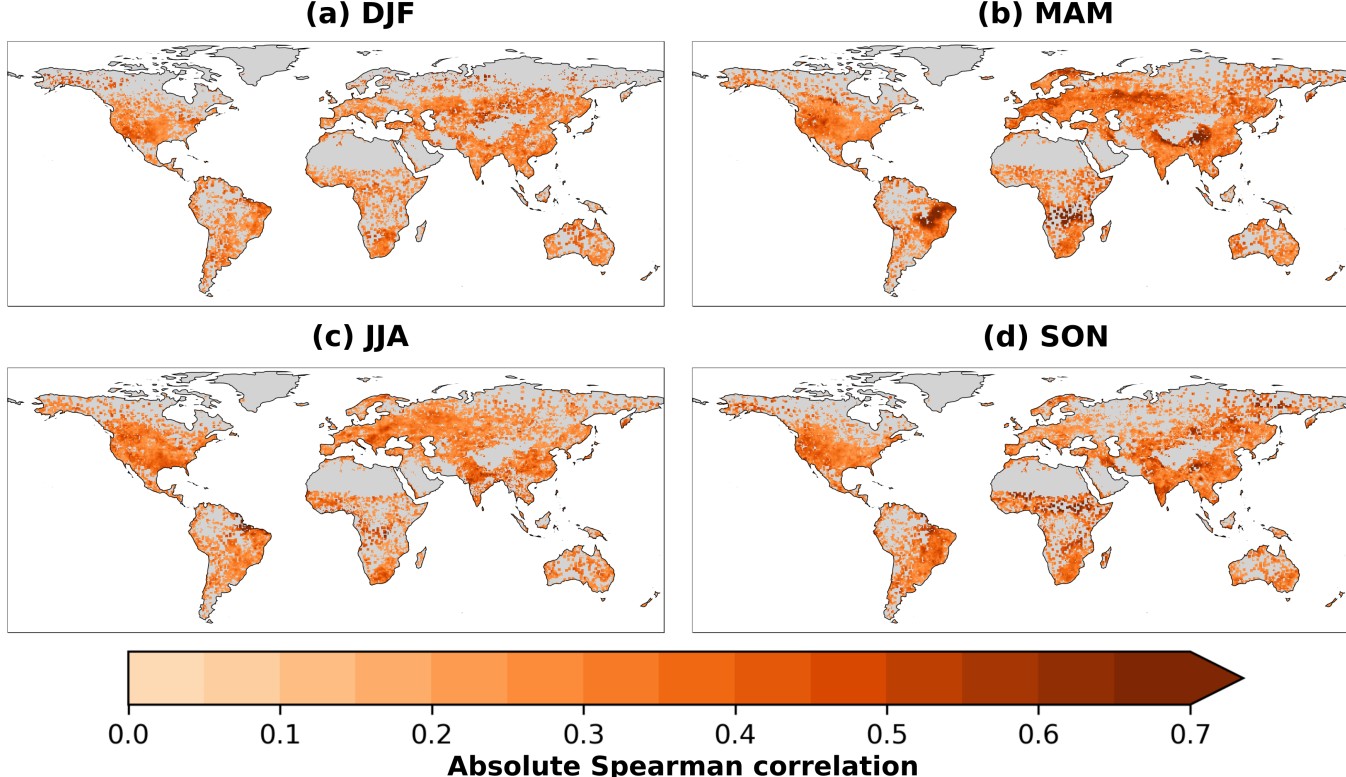

**Figure 4.** Strongest absolute Spearman correlation coefficient between the Earth system variables in Table 1 and the temperature forecast error. The values correspond to the respective most relevant individual variables shown in Fig. A4. Only correlations that are significant after correction for multiple testing are considered.

Next, we focus separately on the relevance of each group of variables for temperature forecast errors, as this may be hid-
den behind the most relevant group/variable but still exhibit significant correlations. The climate group is not only the most important group from a global perspective (Fig. 3), but it also shows significant correlations in many areas where it is not top–ranked (Fig. 5). Land surface variables are globally relevant for forecast errors in similar areas as the climate group. The partly coinciding relevance of the climate and land surface groups is related to the strong coupling between them, as for example solar radiation or precipitation are related to soil moisture, vegetation state and evapotranspiration. The circulation group is
still found to be of minor relevance at the global scale. We attribute this to the short time scale variability of the circulation variables such as the winds and the surface pressure differences.

### 3.3   Possible physical mechanisms behind forecast errors in focus regions

Within the six focus regions highlighted in Fig. 3, the relationships between forecast errors and its most relevant predictors show strong seasonal variations (Fig. 6). In the AF focus region, the most relevant variable is specific humidity during the onset



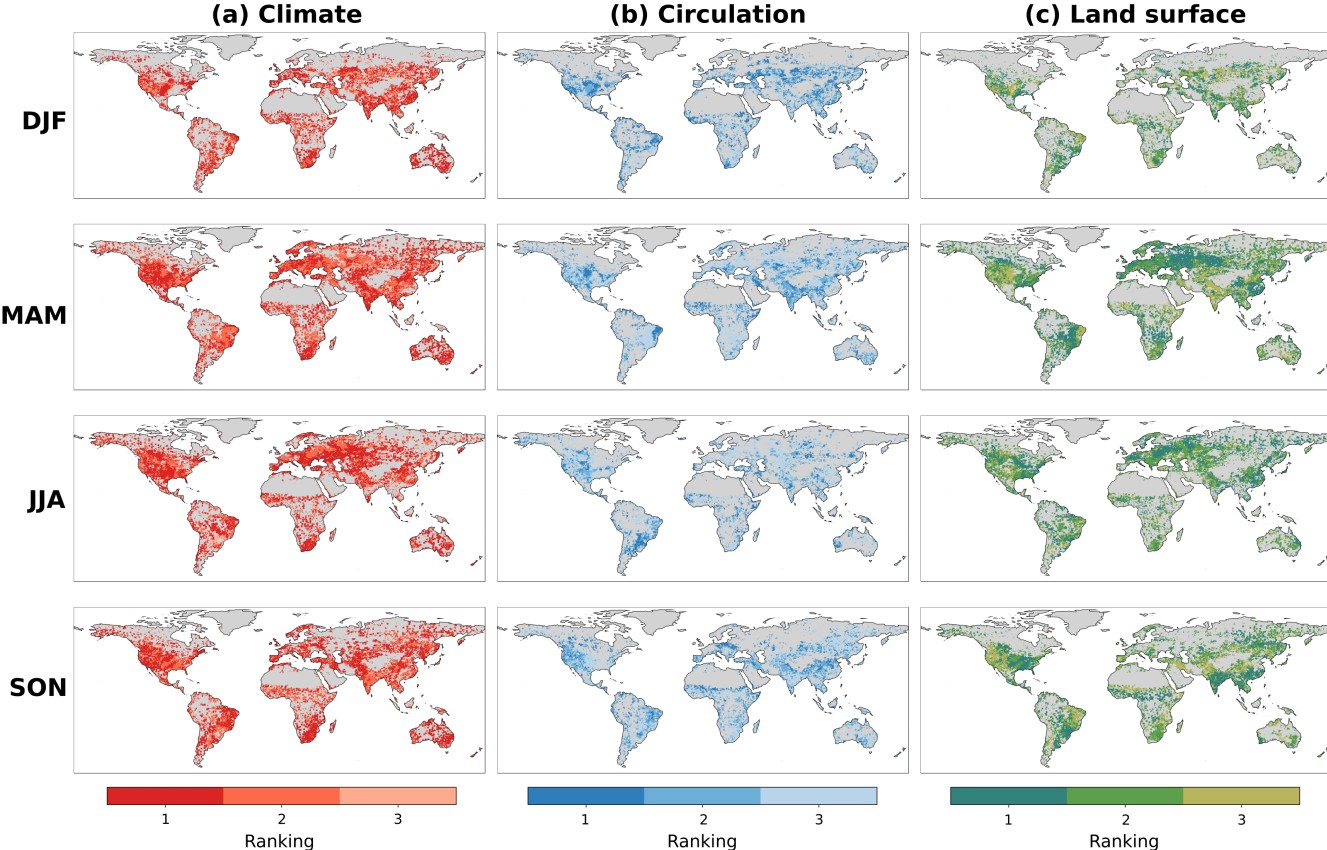

**Figure 5.** Ranking of groups of Earth system variables (climate, circulation, and land surface) according to the Spearman correlation coefficient of the most relevant variable within each group. Gray areas indicate that no variable from the group exhibits a significant correlation with temperature forecast errors.

and development of the regional warm season (SON, DJF). We mainly find errors close to zero around low specific humidity values and increasing errors magnitude with increasing specific humidity. Incorrect initial humidity status can lead to forecast errors in temperature through multiple ways: (*i*) the potential for cloud formation and related reduced incoming solar radiation depends on the amount of water vapor in the air, (*ii*) evapotranspiration is largely determined by convergence of water vapor, and (*iii*) water vapor content in the air affects plant's stomatal resistance and therefore evaporative cooling. We attribute these temperature forecast errors to an inadequate representation of these three processes in the forecasting system.

The most relevant variable in the AS focus region for most seasons is evaporative fraction anomaly. For negative (positive) values of this variable, we see overestimation (underestimation) of temperature. One possible reason is the outdated representation of the low vegetation cover in HTESSEL around this region, as found in Boussetta et al. (2021). An inaccurate prescription of vegetation type and cover fraction can lead to a misrepresented evapotranspiration response resulting in temperature errors.



**Figure 6.** Relationships between the most relevant Earth system variable and forecast error in the focus regions of Africa, Asia, Australia, Europe, North America, and South America (top to bottom) introduced in Table 2 and Fig. 3. Relationships are shown for all seasons as smoothing lines (loess filter) fitted to the underlying point data; each line represents the relationship at one grid pixel. Light–colored smoothing lines indicate that the variable is not the most relevant in the respective season and is only shown for completeness.





In the AU focus region, the most relevant variable is precipitation. AU is a semi–arid focus region, therefore it is expected that precipitation events affect air temperature. We mainly see the influence of precipitation on temperature forecast errors after a threshold around 10 mm/d, and generally the related errors are positive. One pathway in which precipitation influences temperature after three weeks lead time is through the infiltration in the soil and the subsequent evaporative cooling (Miralles et al., 2012; Orth and Seneviratne, 2014). We attribute this association between large precipitation values and positive tempera-

ture forecast errors with the misrepresentation of the evaporative cooling resulting from elevated (surface) soil moisture in this region.

        In the EU focus region, the surface soil moisture is the most relevant variable. During MAM and JJA, there is temperature overestimation (underestimation) in the presence of low (high) soil moisture content. This region is located in a transitional climate regime (between water and energy–controlled conditions) where the land–atmosphere coupling is typically strong

(Seneviratne et al., 2010; Orth, 2021). The strong variability between the smoothing lines in this region in Fig. 6 reflects the heterogeneous land surface with variable soil and vegetation characteristics. This is particularly difficult to represent in the forecasting model and similar to AS, the EU focus region's vegetation representation in HTESSEL is outdated, for both low and high vegetation cover (Boussetta et al., 2021), therefore we attribute the high correlation between temperature forecast errors and soil moisture to the misrepresentation of the land–atmosphere coupling.

For both NA and SA focus regions, the most relevant variables are related to zonal and meridional surface pressure difference anomalies, respectively. These processes drive wind's magnitude and direction and also moisture and heat flux transport. The representation of winds in these regions might be affected also by the high biases in the vegetation cover in HTESSEL which include potentially erroneous assumptions on surface roughness. Even though these processes mainly occur over short time scales, there are cases in which a blocking regime can last for several days and affect temperature forecasts (Grams et al.,

250     2018).

### 3.4    What is the potential to improve temperature forecasts at the S2S time scale?

We identify in Fig. 7 regions where (*i*) forecast errors are relatively high and (*ii*) rather well explained with any of the considered Earth system variables, which indicates some potential for improvement. We use the quantile criteria described in Sect. 2.5. Regions that exceed the 90th percentile in both tested conditions have the highest potential for improvement. We find the most

promising regions around the northern hemisphere extratropics, especially in western North America, central Europe and in the eastern Himalayas. We also see medium potential over some regions in central Africa and central and eastern South America during DJF and JJA. These regions display a climate variable as the most relevant Earth system variable. This could be due to (*i*) deficiencies in the assimilation of climate variables prior to forecast initialization and (*ii*) imperfect process representations in the forecasting model which limits its ability to accurately propagate the initial climate state into the subsequent weeks.

In the case of the climate variables, the detected potential for improving temperature forecasts is probably related to sparse precipitation observations available for assimilation, as well as an imperfect representation of the coupling of precipitation and radiation with temperature in the model. The latter might be more relevant in Europe and North America with transitional




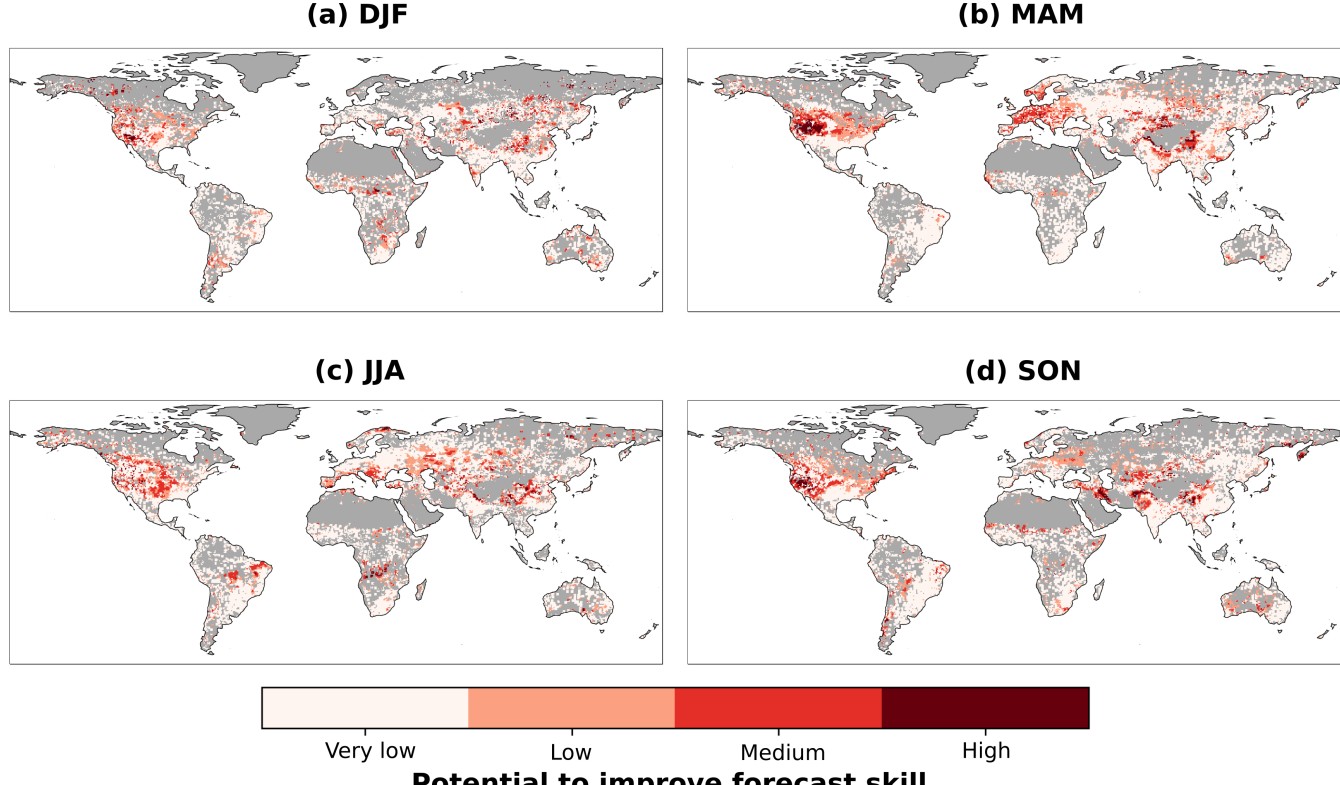

**Figure 7.** Potential for improvement in temperature forecast skill according to magnitude of forecast error from Fig. 2 and mean correlation value from Fig. 4.

conditions between energy and water–controlled evapotranspiration, while additional precipitation information could probably improve forecasts around the Himalayas and in some regions in Africa.

In contrast, we find relatively low potential mainly across eastern Europe, Australia and India. This low potential is probably related to low forecast errors (Fig. 2) and not necessarily low correlation values (Fig. 4).

## 4    Conclusions

We analyze temperature forecast errors in subseasonal forecasts of the ECMWF and their relationship with a set of Earth system variables. Thereby we compute correlations between forecast errors and Earth system variables as a measure of their
potential to inform even more accurate temperature forecasts. While previous studies have assessed the role of individual (sets of) variables, we move beyond the state–of–the–art by analyzing a comprehensive set of Earth system variables covering multiple components of this system including several potential sources of sub–seasonal forecast skill.





The results show that climate–related variables, particularly precipitation and surface solar radiation, are globally the most relevant variables from our considered set of Earth system variables. There is still room for improvement in terms of climate data assimilation in data–sparse regions, and in terms of the modeled evolution of the climate over the forecasting period through, for instance, continuously improving descriptions of e.g. atmospheric dynamics, convection and condensation schemes.

Next to this, the land surface–related variables, particularly surface soil moisture, are also important in many regions. This is specially relevant for subseasonal forecasts given the profound memory characteristics of land surface variables for which anomalies present at the forecast start can persist for weeks or months. Although to a smaller extent than for the climate predictors, we find regions and seasons in which the land surface information is strongly correlated with the temperature forecast errors. This also emphasizes the importance of accounting for the land surface variability, taking advantage of the recent developments in in–situ observations and satellite based data (Balsamo et al., 2018; Eyre et al., 2022). For instance, regions with water–limited conditions (typically semi–arid regions) can experience impacts of soil moisture variability on temperature (Ford et al., 2018; Jach et al., 2022). Moreover, the vegetation (in terms of biomass, greenness and photosynthetic activity) can amplify the land surface effect on temperature through evaporative cooling via the LAI and stomatal resistance. This is particularly relevant at the S2S scale after extreme events like droughts and heatwaves (Bastos et al., 2020; Byrne, 2021) in regions with strong land–atmosphere coupling where vegetation functioning can be affected even after the event is over because of legacy effects such as hydraulic damage or depleted carbon reserves.

The circulation–related variables, even though they are globally the least relevant variables, also highlight regions where temperature responds to large scale circulation patterns and surface pressure regimes. These results align with recent literature on subseasonal forecasts which contains a relatively strong focus on ocean phenomena, such as NAO, MJO and ENSO (Scaife et al., 2016; Vitart and Robertson, 2018; Mariotti et al., 2020; Falkena et al., 2022; Smith et al., 2020; Kim et al., 2021; Liu et al., 2021; Meehl et al., 2021). These phenomena are mainly driven by surface pressure differences and sea surface temperatures (Ehsan et al., 2021). An improved representation of these variables therefore offers the potential to enhance forecast skill in some regions of the world.

A main limitation in our analysis approach is that the considered Earth system variables are correlated with each other, and that interactions between them are not taken into account with the individual correlations we compute. However, our focus is on first–order effects which can be captured with our approach. Further, we note that the reported correlations, and the rankings between them, are not solely related to physical linkages and respective deficiencies in the forecast model and data assimilation system, but can also be affected by different levels of precision of the observational data sources. This means that noisy observation–based records might not correlate as strongly with temperature forecast errors as they would if they were measured more accurately. Therefore, our analysis reflects the current potential of each Earth system variable to inform more accurate temperature forecasts, rather than the full potential which might be exploited by even more accurate future Earth observations. Finally, our results are limited also by our considered set of Earth system variables; even though we use a





comprehensive set of variables, there might be variables or states which are not yet observed at large spatial scales but relevant for improving the forecast model and its forecasts.

Despite the limitations of this study, our findings do provide information on processes that can be improved to increase the temperature forecast skill of the ECMWF–S2S forecasting system. Accurate predictions at this time scale are particularly im-
310   portant for issuing early warnings of extreme events, for adequately managing resources, and, more importantly, for minimizing human risks (White et al., 2017; Vitart and Robertson, 2018).





## Appendix A: Supplementary methods and results

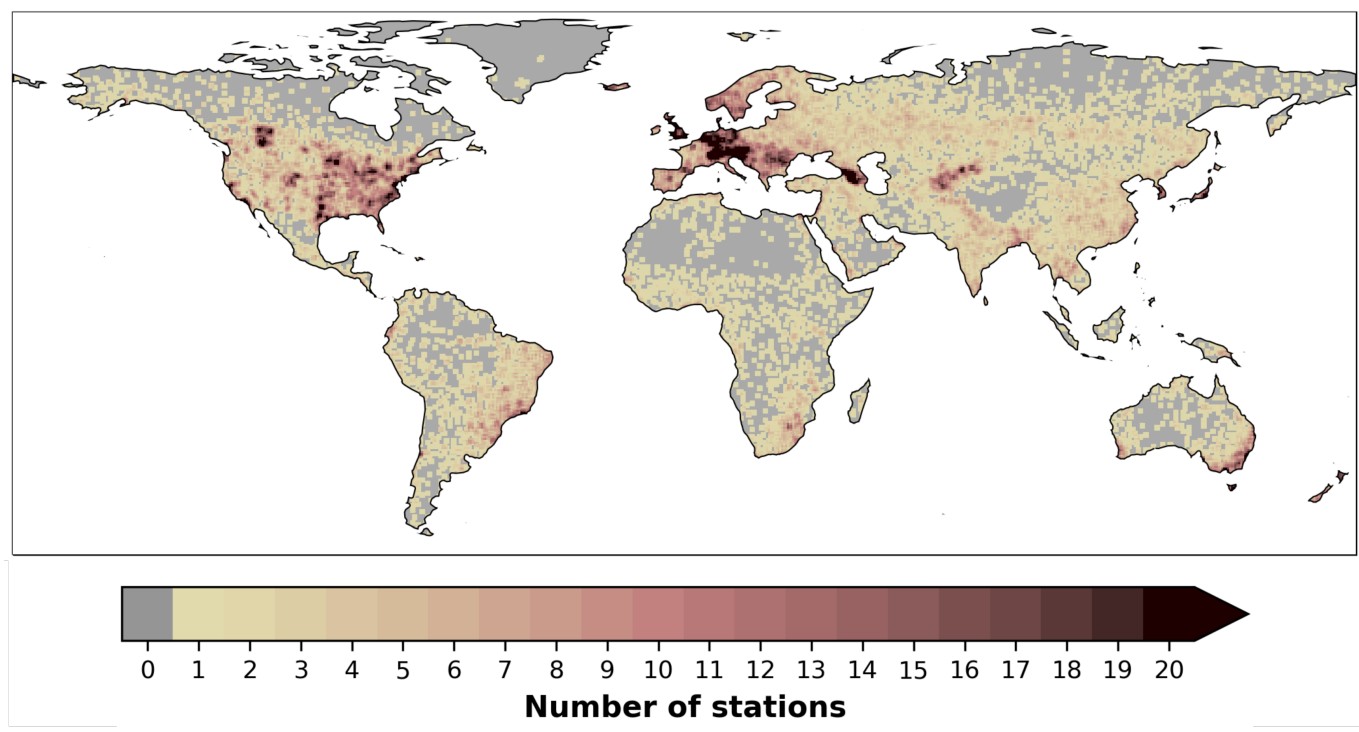

**Figure A1.** Sum of the temperature stations located in each grid cell and its 8 neighboring grid cells. This map is used to mask out regions with no temperature observations. Data available at https://ftp.cpc.ncep.noaa.gov/cadb_v2/library/.





**Figure A2.** Heatmaps of forecast error according to the number of CPC stations around the grid cells (X axis) and the quartiles of standard deviation of altitude in the grid cells (Y axis).





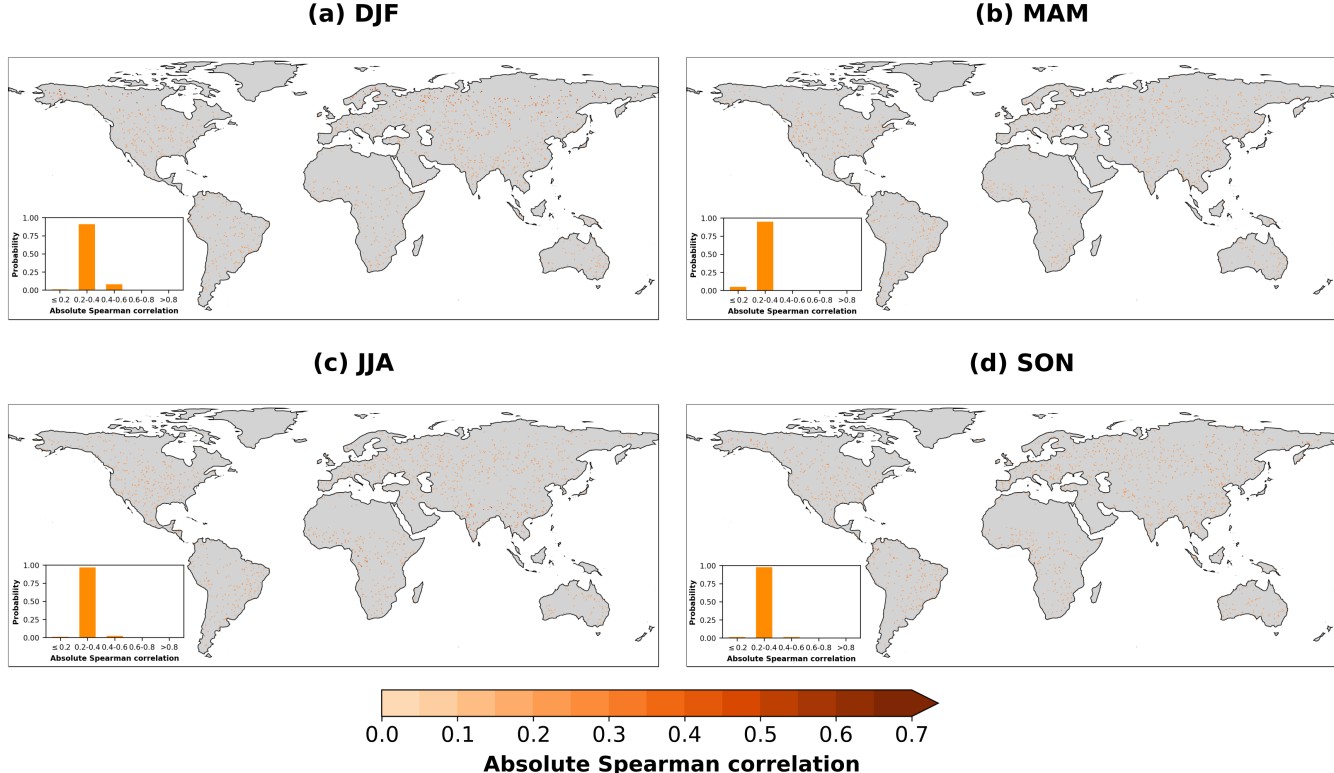

**Figure A3.** Strongest absolute Spearman correlation coefficient between the Earth system variables in Table 1 randomly permuted and the temperature forecast error in Fig. 2. The inset barplots represent the histograms of these global Spearman correlation values.



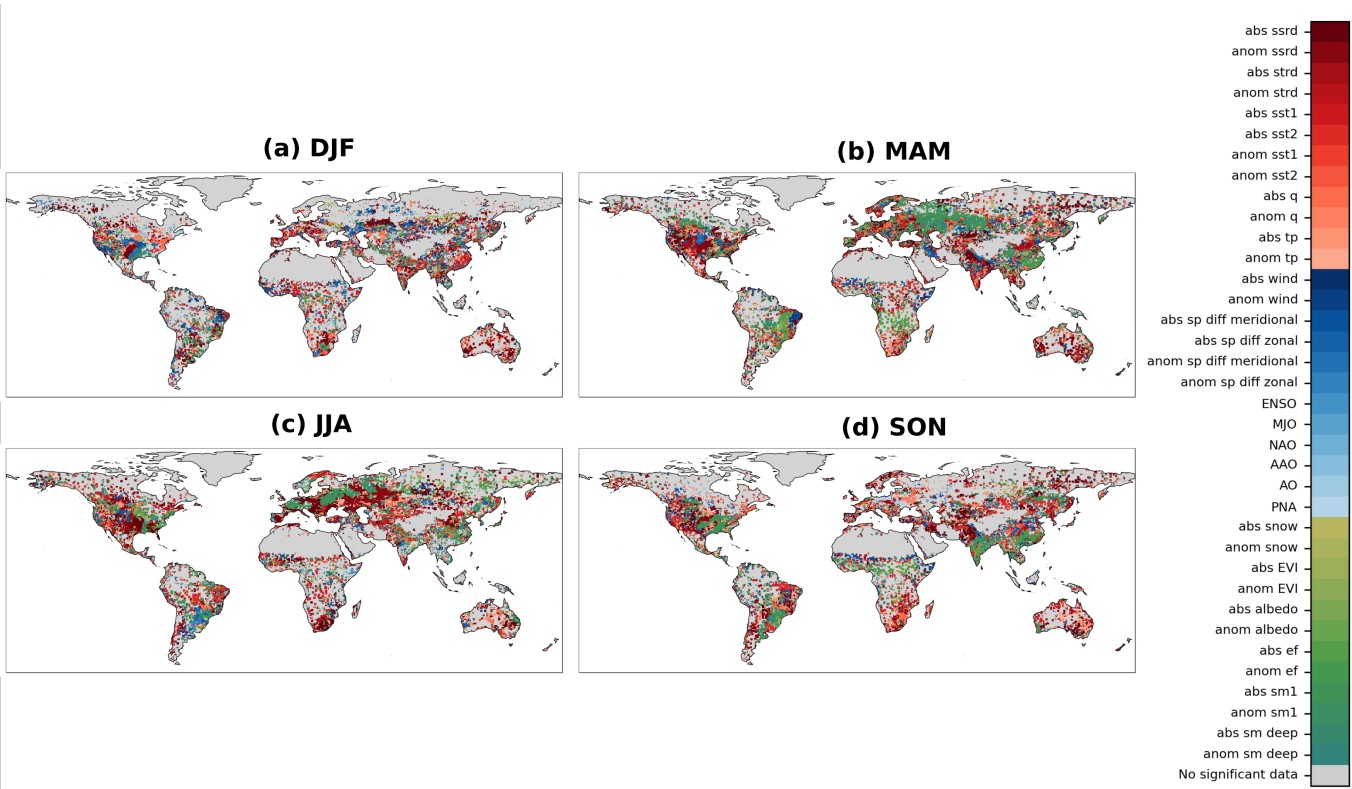

**Figure A4.** Seasonal cycle of each grid cell's most relevant Earth system variable in terms of its relationship with forecast error. The three groups of colors represent the three groups of Earth system variables: climate (red), circulation (blue), land surface (green). Grid cells with no significant Spearman correlation are colored in gray. A NetCDF file with this data is provided in Supplementary material.




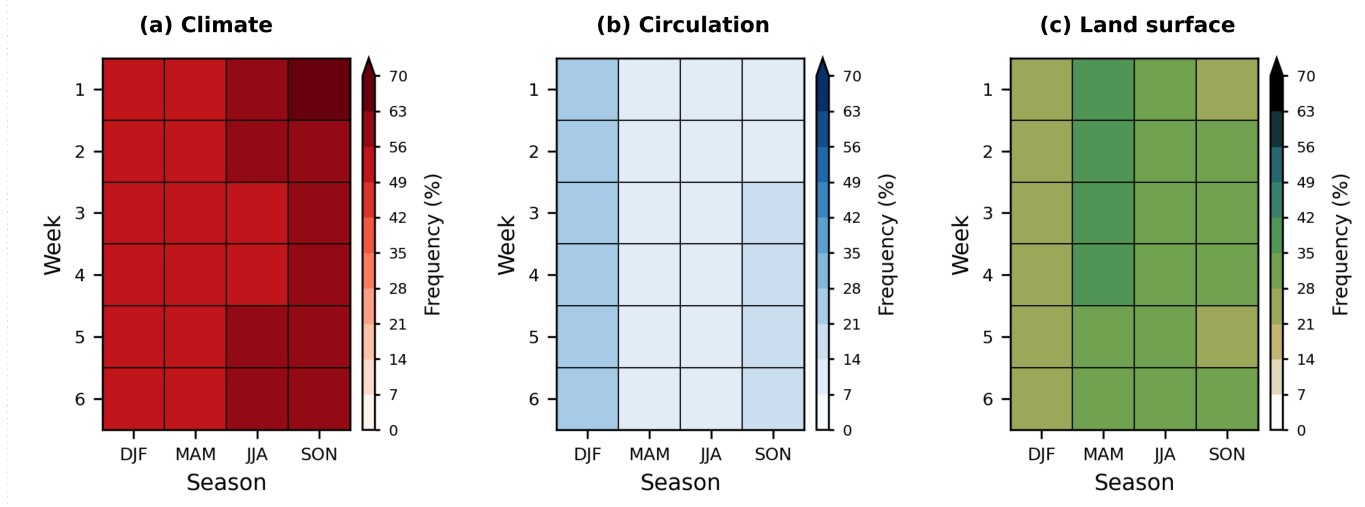

**Figure A5.** Fraction of grid cells (%) where each group of Earth system variables is the most relevant across lead times (Y axis) for each season (X axis).



*Code and data availability.* The S2S forecast temperature data are available at https://apps.ecmwf.int/datasets/data/s2s/. The CPC reference temperature data are available at https://psl.noaa.gov/data/gridded/data.cpc.globaltemp.html. The variables from ERA5 are available at https://cds.climate.copernicus.eu/. The precipitation data from GPCP are available at https://www.ncei.noaa.gov/products/climate-data-records/precipitation-gpcp-daily. The El Niño 3.4 index is available at https://climexp.knmi.nl/selectdailyindex.cgi?id=someone@somewhere. The MJO data is avialable at http://www.bom.gov.au/climate/mjo/. The NAO, PNA, AAO and AO circulation indices data from NOAA are available at https://ftp.cpc.ncep.noaa.gov/cwlinks/. The EVI and albedo data from MODIS are available through NASA's data catalogue at https://lpdaac.usgs.gov/products/mod13c1v006/ and https://lpdaac.usgs.gov/products/mcd43d42v006/, respectively. The snow data from MODIS are available through the National Snow & Ice Data Center catalogue at https://nsidc.org/data/MOD10A1/versions/61/. Both the evaporative fraction data from FLUXCOM and the soil moisture data from SoMo.ml are available at the Data Portal of the Max Planck Institute for Biogeochemistry at https://www.bgc-jena.mpg.de/geodb/projects/Data.php.

*Author contributions.* MRV, SO, AlexB and RO jointly designed the study. MRV performed the computations and data analysis. UW carried out data acquisition and formatting. MRV, SO, AlexB, RDK, GB, GA, AnaB, MR and RO contributed to the writing of the paper and the discussion and interpretation of the results

*Competing interests.* The authors declare that they have no conflict of interest

*Acknowledgements.* Melissa Ruiz–Vásquez acknowledges support from the International Max Planck Research School for Global Biogeochemical Cycles (IMPRS). René Orth was supported by the German Research Foundation (Emmy Noether Grant 391059971). Sungmin O acknowledges the Brain Pool program funded by the Ministry of Science and ICT through the National Research Foundation of Korea (Grant NRF-2021H1D3A2A02040136). We thank Frederic Vitart for his guidance with the S2S data acquisition and the Hydrosphere–Biosphere–Climate Interactions group in the Biogeochemical Integration Department of the Max Planck Institute for Biogeochemistry for fruitful discussions that have contributed to the interpretation of the results and the design of the figures.



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
