# Peer review of "Exploring the relationship between temperature forecast errors and Earth system variables"

_EGUsphere, 2022_

## Author Response (AR1)

**Point by point reply to comments**

We are delighted to read that the reviewers appreciate the content and presentation of our study. We are thankful for their comments and suggestions. Here we present a point by point reply to the two reviewers' comments and to one additional question from the editor.

**First review**

**General assessment :**

This manuscript investigates the main sources of temperature subseasonal forecast skill at the global scale. It evaluates the relationships between potential drivers of three kinds (climate, circulation and land surface) derived from multiple observational/reanalysis datasets, and subsequent temperature forecast errors of the ECMWF extended range reforecasts, at different seasons. Overall, climate drivers tend to prevail, but land surface drivers also greatly contribute to forecast errors. Circulation drivers seem less relevant although not everywhere. Finally, based on correlation strength and forecast error amplitude, the authors highlight regions where subseasonal forecast skill could be potentially improved across seasons.

The scope of this study is both original and of great interest for the S2S community. The underlying rationale is relatively simple, but relevant too, so that I consider it an asset here. I would also like to stress that the paper is well written, well articulated, clear and enjoyable to read. My one main concern is about the evaluation of forecast errors. I feel like the metric used is not well suited for such a study where a distinction is made between seasons (see details below). It may have a limited impact in terms of the Spearman correlations found, but probably not on those of section 3.4. I think the authors should consider either correcting this metric or justify its relevance in the light of my comment below.

We appreciate the effort and time devoted by the reviewer in writing such constructive remarks. We think that the quality of our manuscript has significantly improved after addressing the reviewer's comments.

**Main comment:**

L. 150-151: I have the feeling that smaller errors found in transition seasons could also be due to the metric used here. This metric is based on departures from annual average temperature (i.e. no annual cycle). Therefore, for a given year, over mid and high latitudes at least, the annual average must be relatively close to fall and spring average temperatures, but substantially higher than winter and lower than summer average temperatures. Consequently, summer and winter forecast departures from annual average temperature must be generally higher (in absolute value) than fall and spring counterparts, both for forecasts and observations. Finally the amplitude range of the resulting forecast errors is probably larger as well. I am puzzled by this choice of annual average temperature to compute departures here. Why not use weekly (or at least monthly) climatologies based on

the 2001-2017 period instead? This would have avoided the issue of seasonality and that of changing the reference every year.

A1. We thank the reviewer for this relevant comment. We introduced the modification of the forecast metric (Eq. 1) proposed by the reviewer: we use weekly climatologies (week-of-year averages) based on the 2001-2017 period instead of the annual averages. This metric is explained in lines 100-108 of the revised manuscript.

See in the image below the weekly climatologies for 4 example grid cells, for both reference and model datasets. Our main findings are not substantially affected by the adapted calculation of the forecast errors, maybe because the weekly climatologies between both datasets are similar.

[Figure]

Two more minor comments on this metric or its interpretation are reported below.

**Minor points:**

L.66-67: To what extent could the use of 2 model versions affect your results? I understand that initialization is unchanged, but readers not aware of the changes between these 2 model versions might wonder if they concern, say, a land/vegetation scheme or/and a key atmospheric parameterization for example. Such changes are prone to modify the

relationships studied in this manuscript. If possible, I would suggest defending this particular point.

A2. According to the ECMWF model description (available here: https://confluence.ecmwf.int/display/S2S/ECMWF+model+description), there are no differences in the 2 model versions used in this study that can affect our results from the S2S dataset. The version that introduces differences with respect to the CY46R1 in the S2S dataset is the Cy47R2 which includes more model levels, but our computations are done before this version is implemented. We added some of this information in lines 68-71 of the revised manuscript.

L.72: across S2S literature, the definition of leadtime weeks vary: some studies exclude days 1 to 4 after initialization, so that week 1 is defined as the day-5 to day-11 window (e.g. Vitart 2004, de Andrade et al. 2021). Could you specify - and discuss if need be - your method in this respect ?

A3. We use the first 42 days since forecast initialization in our analysis. As we compute weekly averages we use seven days for each week as follows:

- Week 1: day 1 to 7
- Week 2: day 8 to 14
- Week 3: day 15 to 21
- Week 4: day 22 to 28
- Week 5: day 29 to 35
- Week 6: day 36 to 42

We added this explanation in lines 75-76 of the revised manuscript. Due to the variations in the literature about the definition of lead time weeks and that some studies exclude the first days after initialization, we focus on the week 3. Even though we do not describe in detail our results for every week after forecast initialization we show in Figure A4 in the revised manuscript how our results differ between each analysed week. As it can be seen from that figure, the results do not vary strongly with the lead time that we choose.

L.100: I am not sure how $T_{for}$ (annual average) is computed. If I understand well, you have two 6-week forecasts per week, i.e. 104 forecasts x 6 weeks per year. Not to mention the ensemble members. How do you proceed to compute the forecast annual average temperature and ensure it is comparable with observational average (one realization, by essence)? I would recommend to be more specific, and also to state somewhere in the manuscript that forecasts are actually ensembles, and how these ensembles are handled here. I guess you have been dealing with ensemble means, but this needs to be specified somewhere.

A4. As mentioned before in the response to the main comment, now we use a modified forecast error metric based on week-of-year averages instead of the annual averages. Also, we only use one forecast per week, instead of two. We compute these week-of-year averages as follows:

1. We bin all the daily data according to the week lead time they belong to from week 1 to week 6. The next steps are computed independently for every week lead time
2. We compute weekly averages from the daily data
3. We compute the multiannual average of the same week-of-year

About the comment of the ensembles, we use the S2S reforecasts produced only by the ECMWF. These are global ensembles that simulate *i*) initial uncertainties using singular vectors and ensembles of data assimilation and *ii*) model uncertainties due to physical parameterizations using a stochastic scheme. The ensemble is based on 51 members and we use the average of this ensemble that is available in the S2S dataportal (https://apps.ecmwf.int/datasets/data/s2s-reforecasts-daily-averaged-ecmf/levtype=sfc/type=cf/).

We included these points in lines 65-67 and 100-108 of the revised manuscript.

L. 156-158: Agreed, but could it also be related to the greater temperature variability over mid-latitudes? I mean that if you would compute the seasonal average of $(T_{i,for} - T_{for})$ absolute values, for each grid cell, I expect these values to be lower at low latitudes, and therefore, the forecast errors end up lower as well (see e.g. Extended Data Fig. 7 and 8 in Tamarin-Brodsky et al. (2020)). I am not 100% affirmative but since you have not normalized temperature anomalies with their standard deviation, this could explain some (most?) of the meridional gradient depicted in Fig. 2.

A5. We agree that forecast errors in the extratropics might be higher than in the tropics due to a higher temperature variability. We added this argument in lines 165-166 of the revised manuscript.

Table 1 layout: I would suggest to make it clearer that when the column "Source" is empty, it means "similar source as above". Maybe a double quote could do ? And also, if possible and if allowed by the editor, try to reduce the font to have less line breaks. This would ease the reading.

A6. We modified the structure of Table 1 and it now has less line breaks. We also centered the cells in 'Source' and 'Reference' columns when they are shared by multiple rows.

Figure 6, NA region, DJF season: by eye, significant correlation seems quite unlikely although this may be due to a "Pearson correlation oriented" perception instead of Spearman. Why not apply a lighter color shade, or a dashed style for instance, to the smoothing lines corresponding to pixels without significant correlation? Alternatively, you could indicate in the subplots the percentage of pixels of each region with significant correlation.

A7. We included the percentage of grid cells in each panel that have a significant correlation between forecast error and the Earth system variable depicted.

L.159: typo: parenthesis issue

A8. It was adapted.

L.292: I am not sure it is correct to describe NAO and MJO as "ocean phenomena"

A9. These two phenomena are part oceanic, part atmospheric phenomena. We rephrase this sentence as weather phenomena in line 325 of the revised manuscript.

Figure 6 : last row (SA region): the x-axis tick labels are arguably wrong (no positive values)

A10. This row now depicts a different Earth system variable.

Another question that comes to mind when reading your conclusion is the extent to which the same Earth system variables would contribute to explain temperature forecast errors in the same regions for other S2S forecast systems. For example the spatial patterns of subseasonal temperature forecast skill show similarities between models and some predictability drivers are known to impact the same regions for different models (e.g. Ardilouze et al. 2021). I understand this would go way beyond the scope of this study, but I mention it for consideration.

A11. This is a very interesting point. As mentioned by the reviewer, this point is beyond the scope of our study, but still it is a potential follow-up analysis. We can only speculate that other forecasting models would exhibit similar patterns in sources of predictability when using the same drivers that we explore here. In addition, machine-learning-based studies in temperature forecasts at the subseasonal scale have highlighted some of the same drivers found here as important sources of predictability (Rasp et al., 2020; Herman and Schumacher, 2018). We included a discussion of this point in lines 343-350 of the revised manuscript.

Rasp, S., Dueben, P. D., Scher, S., Weyn, J. A., Mouatadid, S., & Thuerey, N. (2020). WeatherBench: A benchmark data set for data-driven weather forecasting. Journal of Advances in Modeling Earth Systems, 12, e2020MS002203. https://doi.org/10.1029/2020MS002203

Herman, G. R., & Schumacher, R. S. (2018). Money Doesn't Grow on Trees, but Forecasts Do: Forecasting Extreme Precipitation with Random Forests, *Monthly Weather Review*, *146*(5), 1571-1600. https://doi.org/10.1175/MWR-D-17-0250.1

**References:**

Tamarin-Brodsky, T., Hodges, K., Hoskins, B.J. *et al.* : Changes in Northern Hemisphere temperature variability shaped by regional warming patterns. Nat. Geosci. 13**,** 414–421, 2020

Vitart, F.: Monthly forecasting at ECMWF, Mon. Weather Rev., 132, 2761–2779, 2004

de Andrade, F. M., Young, M. P., MacLeod, D., Hirons, L. C., Woolnough, S. J., and Black, E.: Subseasonal Precipitation Prediction for Africa: Forecast Evaluation and Sources of Predictability, Weather Forecast., 36, 265–284, 2021

Ardilouze, C., Specq, D., Batté, L., and Cassou, C.: Flow dependence of wintertime subseasonal prediction skill over Europe, Weather Clim. Dynam., 2, 1033–1049, 2021

**Second review**

**General Comments**

In this study, the authors have investigated the relative contribution of observation based ecological, hydrological and meteorological variables in explaining weekly temperature forecast errors in the ECMWF Subseasonal to Seasonal (S2S) reforecasts during 2000-2017, using lead times of 1-6 weeks. Temperature forecast errors are found to be most strongly affected by climate related variables such as surface solar radiation and precipitation. However, vegetation greenness and soil moisture are found to be relevant for central Europe, eastern North America and southeastern Asia. Authors claim that the relationships between forecast errors and independent Earth observations reveal new variables on which future forecasting system development could focus by considering related process representations in detail and data assimilation, to improve subseasonal to seasonal forecasts.

The paper is well-written, lucid and enjoyable and could be a valuable contribution to subseasonal to seasonal scale research. I have a few comments which may be noted below.

We appreciate the positive comments and constructive feedback from the reviewer. We think that the quality of our manuscript has significantly improved after addressing the reviewer's comments.

**Specific comments**

- Why was annual mean temperature considered in the computation of the metric? It is possible that the forecasts and the observations have different annual cycles in a year as well as different interannual variability; so that would add additional biases while computing the weekly forecast errors.

  B1. We thank the reviewer for raising this point, which was also mentioned by reviewer #1. In response, we introduced a modification in the forecast error metric: we use weekly climatologies based on the 2001-2017 period instead of the annual averages to account for potentially different seasonal cycles in forecasts and observations. This metric (Eq. 1) is explained in lines 100-108 of the revised manuscript.

- The areas in Figure 3 either have proximity to the ocean or are inland, and the results based on these small areas have been generalized for these six regions. I wonder, how much does the position of the selected areas affect the relative influence of climate, circulation and land surface variables on temperature forecast errors?

  B2. The selected regions (black squares in Figure 3) are case studies to further understand the relationships between the forecast errors and the most relevant variable within those regions; we chose the regions based on the most relevant variables, not based on their geographical locations. We expect a strong influence of the positions of the regions in our results of the most relevant Earth system variables, therefore we do not want to imply with our results that the results can be generalised to the whole continent where each region is located. We apologise for the confusion; To avoid any confusion, we changed the name of the regions following the variable of interest instead of the continent they are located in. The new focus regions are: specific humidity region (sq), evaporative fraction region (ef), total precipitation region (tp), surface soil moisture region (sm surf), meridional surface pressure differences region (meridional sp) and Enso region (enso). They are introduced in Section 2.4 of the revised manuscript.

- If we compare Figure A4 and the global "hot-spots" of land-atmosphere coupling (Koster et al. 2004), it is surprising that in JJA, surface and deep layer soil moisture do not turn out to be the most relevant Earth system variables for temperature forecast errors over Africa, NA and India. Rather, climate and circulation related variables appear to have a greater impact. I wonder if this is due to some deficiencies in the land surface scheme, land-atmosphere coupling or some other factor?

  B3. This is a very good point. We can expect that during this season of strong land-atmosphere coupling in these regions, soil moisture variables would be particularly relevant for forecast errors. While not shown in Figure A3 of the revised manuscript as the most relevant variable, the land surface variables are still related to temperature forecast errors in JJA across Africa, North America and India even though not always as the most relevant variable (Figure 5). Furthermore, if soil moisture is not the most relevant driver for forecast errors in those regions during JJA, it might be because the land surface scheme is either very good (and it can not be further improved) or the climate and circulation variables are comparatively less well assimilated/represented. This way, we think that other climatic processes related with precipitation, for instance cloud formation and the migration of the intertropical convergence zone (ITCZ) may be more relevant for forecast errors in those regions during the rainy season (JJA). We clarified these points in lines 193-199 of the revised manuscript.

- The memory of surface soil moisture anomalies is much less (except in arid, forested and snow-covered regions) compared to that of the root zone and is certainly lower than the lead time of 3 weeks considered for temperature forecasts. However surface soil moisture turns out to be the relevant variable for temperature forecast errors than deeper layer soil moisture. What is the reason?

B4. Surface soil moisture affects evaporation from soils as well as the transpiration from short vegetation which lacks deep-reaching roots. Through these pathways it can have a significant impact on the surface energy balance and hence temperature. Furthermore, surface soil moisture typically exhibits a larger variability compared with deeper soil moisture which can also lead to stronger impacts on temperature.

In dense forest regions we expect that the deep soil moisture has a stronger effect in temperature forecast errors than the surface soil moisture because their rooting systems are more suitable to extract water from the deepest layers. Nevertheless, as seen in Figures 2 and 3, there are no results in a large fraction of the grid cells located in rainforests probably due to the low density of temperature observations. This can explain why in Table 3 we see a small fraction of grid cells where deep soil moisture is the most relevant driver for forecast errors.

We discussed these points in lines 308-317 in the revised manuscript.

- In the South Asian summer monsoon region the atmosphere depicts significant 'internal' low-frequency variability that could be generated due to various factors such as non-linear scale interactions, the distribution of orography, land and ocean and their interaction with wind flow etc. How much does this factor affect the evolution of temperature forecasts errors over the SA region in JJA?

B5. Did the reviewer mean the AS region instead of the SA region? In that case, we agree with the reviewer that this region is characterised by a complex topography and by the presence of nonlinear processes due to the summer monsoon that make it difficult to predict temperature, especially at the subseasonal level. Besides, and probably because of the complex topography (especially in the northernmost part of the region), there is not enough data assimilation in the forecasting system to accurately constrain the initial conditions of the forecasting model, which further increases the errors. Nevertheless, this region does not particularly show strong forecast errors like other regions (Figures 2 and 7), probably due to a good representation of the monsoon onset and magnitude within the forecasting system, which provides most of the predictability for temperature in this region. We mentioned the role of non-linear interactions in forecast errors in lines 168-171 of the revised manuscript.

**References**

Koster R. D. and Co-authors (2004), Regions of strong coupling between soil moisture and precipitation, *Science*, 305, 1138-1140.

**Question from the editor**

Dear authors

Revisions are necessary according to the referee reports. In addition to the referee comments have one additional question. How do you exclude correlations between the climate and the circulation variables and how would such correlations affect your analyses?

We think this is a very crucial point in our study and we thank the editor for raising the question.

We acknowledge that a main limitation in our analysis approach is that the considered Earth system variables are correlated with each other (especially those from the climate group). As our methodology is based on individual correlations it is not directly affected by relationships among the considered variables which would be the case for, e.g., multivariate approaches. Yet, we note that our results can only indicate first–order effects by pinpointing the most relevant variables and future work should focus on disentangling the joint effects of multiple variables to advance the understanding of temperature forecast errors even further. Also we computed a correlation matrix (Fig. A6 of the revised manuscript) between the considered Earth system variables to account for the magnitude of these cross-correlations and in general it is the climate-related variables that are most affected by co-linearities. We discussed this point in lines 222-225 and 330-342 of the revised manuscript.